

# South Pacific Subtropical High from the late Holocene to the end of the 21st century: insights from climate proxies and general circulation models

Valentina Flores-Aqueveque[1,3], Maisa Rojas[2,3,4], Catalina Aguirre[4,5,6], Paola A. Arias[7], Charles González[1]

[1] Departamento de Geología, Facultad de Ciencias Físicas y Matemáticas, Universidad de Chile, Plaza Ercilla 803, Santiago, Chile
[2] Departamento de Geofísica, Facultad de Ciencias Físicas y Matemáticas, Universidad de Chile, Blanco Encalada 2002, Santiago, Chile
[3] Millennium Nuclei for Paleoclimate
[4] Centro de Ciencia del Clima y la Resiliencia (CR[2])
[5] Escuela de Ingeniería Civil Oceánica, Facultad de Ingeniería, Universidad de Valparaíso, Chile
[6] Centro de Observación Marino para estudios de Riesgos del Ambiente Costero (COSTA-R)
[7] Grupo de Ingeniería y Gestión Ambiental (GIGA), Escuela Ambiental, Facultad de Ingeniería, Universidad de Antioquia, Medellín, Colombia.

*Correspondence to*: Valentina Flores-Aqueveque (vfloresa@uchile.cl)

**Abstract.** The South Pacific Subtropical High (SPSH) is a predominant feature of South American climate. The variability of this high-pressure centre induces changes in the intensity of coastal alongshore winds and precipitation, among others, over southwestern South America. In recent decades, a strengthening and expansion of the SPSH have been observed and attributed to the current global warming. These changes have led to an intensification of the southerly winds along the coast of northern to central Chile, and a decrease in precipitation from central to southern Chile. Motivated by improving our understanding about the regional impacts of climate change in this part of the Southern Hemisphere, we analyze SPSH changes during the two most extreme climate events of the last millennium: the Little Ice Age (LIA) and the Current Warm Period (CWP: 1970-2005), based on paleoclimate records and CMIP5/PMIP3 model simulations. In order to assess the level of agreement of general circulation models, we also compare them with ERA-Interim reanalysis data for the 1979-2009 period as a complementary analysis. Finally, with the aim of evaluating future SPSH behaviour, we include 21th century projections under a RCP8.5 scenario in our analyses. Our results indicate that during the relative warm (cold) period, the SPSH expands (contracts). Together with this change, alongshore winds intensify (weaken) south (north) of ~35ºS; also, Southern Westerly Winds become stronger (weaker) and shift southward (northward). Model results generally underestimate reanalysis data. These changes are in good agreement with paleoclimate records, which suggest that these variations could be related to tropical climate dynamics but also to extratropical phenomena. However, although models adequately represent most of the South American climate changes, they fail in representing the Intertropical Convergence Zone - Hadley Cell system dynamics. Climate model projections indicate that changes recently observed will continue during next decades,





highlighting the need to establish effective mitigation and adaptation strategies against their environmental and socio-economic impacts.

## 1. Introduction

Climate conditions in South America (SA) are the result of the complex interactions among predominant atmospheric
circulation patterns, orographic features, latitudinal differential radiation, and heat and water balances. One of the main features dominating climate in this region, in particular its western boundary, is the South Pacific Subtropical High (SPSH), a quasi-permanent center of high atmospheric pressure developed over the southeastern Pacific Ocean. The present-day behavior of the SPSH is relatively well known. The SPSH varies (intensity and position) at different timescales, from seasonal to interannual, depending on the interplays between forcing from higher and lower latitudes (Grotjahn, 2004) and
the superposition of large-scale phenomena at interannual timescales, such as El Niño-Southern Oscillation (ENSO; Cane, 1998), and the Pacific Decadal Oscillation (PDO; Mantua et al., 1997), at interdecadal timescales (Ancapichún and Garcés-Vargas, 2015). SPSH changes, in turn, influence other elements of the ocean-atmosphere dynamics of SA, such as the meridional winds -and the related upwelling along the coastal central Peru to south-central Chile (e.g., Pizarro et al. 1994; Falvey y Garreaud 2009; Rahn and Garreaud, 2013), and precipitation at the western side of the Andes (e.g., Barrett and
Hameed, 2017).

However, little is known about its past behavior. During last decades, an intensification of coastal southerly winds and a decrease in precipitation have been observed in southwestern SA. These changes have been related to an intensified and a southward shifted SPSH as a response to the current global warming scenario (Falvey and Garreaud, 2009; Rahn and Garreaud, 2013; Ancapichún and Garcés-Vargas, 2015; Schneider et al., 2017; Boisier et al., 2018; Aguirre et al., 2018).
Model projections indicate that this trend will continue during the 21st century for southerly wind intensity (e.g., Garreaud and Falvey, 2009; Belmadani et al., 2014; Aguirre et al., 2018), as well as for precipitation (e.g., Nuñez et al., 2008; Kitoh et al., 2011; Cabré et al., 2016).

Given the fundamental role of the SPSH on the alongshore winds, and therefore over wind-driven upwelling along southeastern Pacific (Croquette et al., 2007; Aguirre et al., 2018) and precipitation (Barrett and Hameed, 2017), the study of
SPSH past variations is essential for understanding its future behavior and key for diagnosing the impacts derived of climate change on the eco-hydrologic dynamics and socio-economic activities in this region, such as fisheries.

In this work, we analyze the SPSH past variations and their influence in the climate of western SA during the last millennium -a period where natural and anthropogenic forcing can be evaluated-, from two lines of evidence: paleoclimate proxies and general circulation models (GCMs). In particular, we focus in two extreme climate periods: the cold Little Ice
Age (LIA) and the Current Warm Period (CWP) in order to contrast the SPSH behavior under different global/regional temperature conditions. Model data for the CWP period are compared with European Centre for Medium-Range Weather Forecasts (ECMWF) ERA-Interim Reanalysis data to evaluate the degree of agreement between GCMs and recent





‘observations’. In addition, with the aim to understand future SPSH conditions and their consequences, we also include 21st century projections under a RCP8.5 scenario in our analyses.

The structure of this paper is organized as follows. First, a climate background of present-day and past changes of SA climate is presented. Section 2 details model and reanalysis data analyzed, and the periods and paleoclimate records

considered. Section 3 presents the results obtained from global model simulations, their comparison with reanalysis data as well as with the information interpreted from paleoclimate records, and the discussion of these evidences. Finally, Section 4 summarizes the main findings and conclusions from this work.

## 1.1  Present-day climate of southwestern South America and future projections

In the Southern Hemisphere (SH), the subsiding branch of the Hadley cell (HC) determines the presence of a quasi-

permanent belt of high surface pressure around 30ºS (e.g., Held and Hou, 1980), whose development over the Pacific Ocean is known as the SPSH. The SPSH extends over much of the Pacific Ocean off the Chilean coast, with a center between 25-30ºS and 90-105°W (Pizarro et al., 1994), and an area of influence exceeding 45ºS during austral summer, when it strengthens (Fuenzalida, 1971). At the northern edge of the SPSH, the air masses flow westward, producing the belt of the trade winds (tropical easterlies). South of the SPSH, a westerly wind belt, known as the Southern Westerly Winds (SWW), is

developed in the mid-latitudes, peaking around 50ºS (Varma et al., 2012). These prevailing winds, that are responsible of the very high regional precipitation on the windward side of the southern Andes (1000–7000 mm yr$^{-1}$, Garreaud et al., 2013), are characterized by a remarkable seasonality driven mainly by changes in sea surface temperature (SST) and atmospheric temperature gradients (Sime et al., 2013). Seasonally, the SPSH is strongest in intensity and has a more poleward position during austral summer, and has a more equatorward position, and less intensity during austral winter, allowing mid-latitude

frontal systems to reach further north, bringing winter rains to central Chile (e.g., Quintana and Aceituno, 2012).

The present-day climatic influence of the SPSH over SA is modulated by large-scale climatic phenomena at interannual to decadal timescales. ENSO (Cane, 1998) dominates global climate variations for the Pacific and the global tropics (Rasmussen and Wallace, 1983) on interannual timescales, ranging quasi-regularly from 3 to 6 years and between 2 to 7 years, according to Dettinger et al. (2000) and Cane (2005), respectively. In southwestern SA, this phenomenon is associated

to higher probability of heavy rainfall in central Chile during winter season, catastrophic flooding in coastal Peru and drought in the Altiplano of Peru and Bolivia (Cane 2005). At decadal timescales (20 to 30 years; Mantua and Hare, 2002), the influence of the PDO (Mantua et al., 1997; Zhang et al., 1997) plays a major role on South American climate. This climate pattern is described as El Niño-like, mainly because its warm (cold) phases are very similar to those of El Niño (La Niña) events, although of smaller amplitude (Garreaud and Battisti, 1999). In SA, the anomalies of rainfall and temperature

related to PDO display a spatial distribution similar to the ENSO-related anomalies (Garreaud et al., 2009).

At higher latitudes, south of 20ºS, the Southern Hemisphere Annular Mode (SAM), also referred as the Antarctic Oscillation (AAO), influences climate variability at intermonthly and interannual timescales (Thompson and Wallace, 2000). The SAM is characterized by two pressure anomalies of opposite signs centered in the Antarctica and a circumglobal band at ~40º-





50ºS, related to the north-south displacement of the SWW (Garreaud et al., 2009). In SA, the influence of this feature is associated with the increase of air temperature between 40º and 60ºS, and a decrease of precipitation in southern Chile during its positive phase, related to a southward shift of the SWW (Thompson et al. 2000; Thompson and Solomon 2002; Marshall et al. 2004; Garreaud et al., 2009).

In addition, due to its outstanding orographic features, the southwestern margin of SA could also be influenced by regional-to-local climate variations that superimpose over these large-scale phenomena (e.g., Rutllant et al., 2003; Moy et al., 2009).

In the past decades, several changes have affected the climate of SA. The most important of these variations are a decrease of precipitation in southwestern SA (Boisier et al., 2016; Boisier et al., 2018) and an intensification of coastal upwelling favorable winds (Schneider et al., 2017) that trigger biological and biogeochemical consequences (Anabalon et al., 2016;

Aguirre et al., 2018). Several authors have associated changes in alongshore winds to an intensified or expanded SPSH (e.g., Falvey and Garreaud, 2009; Belmadani et al., 2014; Ancapichún and Garcés-Vargas, 2015).

These observed trends are also found in numerous model projections for warming scenarios, suggesting that these changes could continue during this century, as result of changes in the SPSH and precipitation, related to a poleward shift and intensification of the SWW (e.g., Fyfe and Saenko, 2006; Ihara and Kushnir, 2009; Chavaillaz et al., 2013; Bracegirdle et al.,

2018). The projection of a poleward displacement of the SPSH that intensifies alongshore winds at the poleward portions of upwelling systems is a clear and robust characteristic at south-central Chile (Rykaczewski et al., 2015). Furthermore, the upwelling system off south-central Chile exhibits a robust future change in timing, intensity and spatial distribution of alongshore winds as consequence of global warming (Belmadani et al., 2014; Wang et al., 2015; Rykaczewski et al., 2015).

**1.2 Late Holocene climate in South America: global events**

The LIA refers to a period of prominent climate anomalies over the past millennium where cold conditions were recorded in most of Europe and parts of North America. It has been established that the LIA developed between 16[th] and mid-19[th] centuries (Mann, 2002), being the last cold period recorded on Earth. However, the timing, magnitude and nature of this event vary significantly from region to region (Bradley and Jones, 1993; Mann et al., 1999) and between archives (Chambers et al., 2014).

Unlike the Northern Hemisphere (NH), where the LIA is well-attested and well-recorded (e.g., PAGES2k Consortium, 2017), in the SH the evidence and detection of this phenomenon in proxy-climate records is less clear with respect to their timing, regional extent and specific associated climatic changes (e.g., Chambers et al., 2014), leading the notion that this event was perhaps not globally synchronous and, therefore, driven by different mechanisms.

In southern SA, evidence of this event has been poorly documented. In general terms, the few reconstructions made suggest

that in this region the LIA was a period characterized by wetter conditions related to changes in the position of the SWW belt (e.g., Lamy et al., 2001; Moy et al., 2008, 2009; Moreno et al., 2014). However, so far there is still not consensus about the trend in temperature during the LIA in this region. Some authors (e.g., Masiokas et al., 2009; Koch and Kilian, 2005; Koch, 2015) have recognized glacier advances in the extratropical SA (17º-55ºS) during this period, which can be interpreted as





cold conditions. However, recently González-Reyes et al., (submitted), based on glacier equilibrium-line altitude modeling, determine that in the Mediterranean Andes (30º-37ºS) there are no marked glacier advances associated to the 'classic' (i.e., NH) LIA. Instead of that, the authors explain that more local factors, such as the Pacific SST variability, would have controlled the glacier mass balance in the region during this period.

Moreover, CMIP5/PMIP3 Last Millennium (LM) simulations have shown weak temperature anomalies during the LM, hindering the identification of this period, especially in the SH (e.g., Rojas et al., 2016; Fig. 1S). This has been associated to the fact that this global event is a response to internal climate variability (PAGES2k Consortium, 2013; Neukom et al., 2014) rather than to external forcing.

On the other hand, an increase in mean global temperature from the late 20[th] century has been observed and highly
documented by several authors (e.g., Jones et al., 1998, 2001; Mann et al., 1999, 2003; Briffa, 2000; Crowley and Lowery, 2000; Folland et al., 2001; Hartmann et al., 2013, among many others). This global warming, also referred as the late-20th Century Warmth or the Current Warm Period (CWP), has been defined as an uniform period of positive temperature trends simultaneous in both hemispheres (e.g., Neukom et al., 2014), with no precedent in the last 1000 years (Mann et al., 1999; Jones et al., 1998, 2001; Crowley and Lowery, 2000; Folland et al., 2001; Jones and Mann, 2004; Marcott et al., 2013;
PAGES2k Consortium, 2017). This characteristic makes the CWP, together with the LIA, the major climate events of the past one or two millennia, a period known as the ''late Holocene'' (Williams and Wigley, 1983).

## 2. Methodology

### 2.1 Climate models for the Last Millennium, the historical period and 21st century projections

In this study, we analyze four CMIP5/PMIP3 climate model experiments (Table 1) for which all three types of simulations
are available: (1) the Last Millennium simulation (past1000; Schmidt et al. 2011, 2012), which considers observed forcing (orbital parameters, solar irradiance, greenhouse gases, land use change and volcanic aerosols) covering the 850-1850 AD period; (2) the historical simulation, which includes natural and anthropogenic (greenhouse gases concentration and aerosols) forcing over the 1850-2005 AD period; and (3) the anthropogenically-forced Representative Concentration Pathways RCP8.5 scenario (named by the equivalent effect in the radiative forcing by 2100 AD, relative to preindustrial
period: +8.5 Wm$^{-2}$) , characterized by increasing greenhouse gas emissions over the period 2006-2100 AD (21C). For each of these, we use the first ensemble member (r1i1p1) of the four models considered.

We particularly analyze different meteorological variables considered to be representative of the main climate components present over western South America. The variables analyzed correspond to monthly mean data for: (a) the SPSH, represented by the sea level pressure (SLP) and the position (latitude, longitude) of its maximum; (b) the intensity of the
meridional winds (e.g. approximately alongshore) at 850 hPa; (c) the SWW defined as the latitudinal position of the maximum zonal wind at 850 hPa, between 40ºS and 60ºS; (d) the Intertropical Convergence Zone (ITCZ) represented by the



position of the maximum value of precipitation between 15ºN and 8ºS; and (e) the Subtropical Jet (STJ), using the latitudinal position of the maximum zonal wind at 200 hPa, north of 48ºS.

Additionally, the global HC is computed over the three periods (LIA, CWP, 21C) using the mean meridional mass stream function.

## 2.2 Global climate events: time periods considered

For the LIA period, we use the spanning time identified in each model by Rojas et al. (2016), based on the annual temperature anomaly over the NH, with respect to the 1000–1850 CE mean, and the meridional SST gradient between the tropical North and South Atlantic (Table 1).

On the other hand, the definition of the onset of the CWP varies regarding the author. Some of them consider it since 1850
CE (Deng et al., 2017) or the entire 20th century (1901-1990) (Briffa et al., 1995; Briffa, 2000; Jones et al., 1998), while other authors refer to 'mid-20th century' (e.g., Crowley and Lowery, 2000; Levitus et al., 2000) or 'late-20th century' (e.g., Mann et al., 2003; Jones and Mann, 2004; Osborn and Briffa, 2006). Recently, Díaz and Vera (2018) considered the CWP in South America as the period between 1951 and 2000 CE, taking into account also the negative rainfall trend observed in the southern Andes.

According to the last IPCC Special Report (Allen et al., 2018), between 1890 and 2010 almost all the current warming could be attributed to human activities, being, in the absence of strong natural forcing, the solar and volcanic contributions accounting for less than ±0.1ºC of the temperature increase between 1890 and 2010. The same report states that since 2000, the human-induced warming accounts for the ~80% of total warming, with the ±20% of uncertainty attributed to solar and volcanic contributions.

Whatever the time period considered, all these studies have led to a consensus regarding that in the past few decades, the mean annual NH temperatures are the warmest of the last 1000 or even 2000 years. For this reason, we consider the CWP as the period covering the years 1970-2000 CE.

Finally, in an analogous way, the RCP8.5 scenario in model projections for the end of the 21st century were considered covering the period between years 2070 and 2100.

## 2.3 Reanalysis data

With the aim of comparing the CMIP5/PMIP3 simulation data for the CWP (historical experiment; 1970-2000) and the present-day climate observational dataset, we use annual and seasonal mean data from ERA-Interim Reanalysis for the period 1979-2009. This project, extensively used in climate research, consists of a global scale dataset that includes recorded
climate observations of atmospheric weather (data available online at https://www.ecmwf.int/en/forecasts/datasets/reanalysis-datasets/era-interim), with a spatial resolution of 0.75º x 0.75º



(latitude x longitude), approximately, of the most relevant meteorological variables (wind, pressure, temperature, cloud cover, among others). Its primary temporal resolution is 3 hours for surface parameters, starting in 1979 and being continuously updated in near-real time (Dee et al., 2011).

We use ERA-Interim Reanalysis because of its high spatial resolution and the continuous improvement in the quality of their data.

### 2.4 Climate records of southwestern South America

In order to test the robustness of CMIP5/PMIP3 LM climate simulations, we carry out an integration of the results obtained from model simulations as well as paleoclimate and paleoenvironmental reconstructions of several regional high-resolution records, mainly from southwestern South America, but also from other regions of the world.

We consider records that: (1) provide information about the main climate features affecting southwestern South America, (2) have a minimum length of 500 years to potentially record the LIA and the CWP, and (3) have a resolution at interannual-to-centennial timescales (Fig. 1).

The selected records fulfill at least three criteria defined in PAGES2k-Network. However, we also include records that do not meet these criteria, but are particularly valuable because of its temporal resolution or location, considering the sparseness

of records in this region. Detail of the selected records and their main characteristics are presented in Table 2.

### 3. Results and discussions

### 3.1 Paleoclimate evidence

Several paleoclimate records analyzed (Table 3, Fig. 1) indicate that important changes have occurred during the late Holocene along the western margin of South America. Salvatecci et al. (2014) analyzed marine laminated sediments off

Pisco (14ºS/76ºW, Perú), interpreting a period of weak alongshore winds related to a contracted SPSH during the LIA and stronger winds due to an expanded anticyclone during the CWP. Recent climatological data seem to support these observations (Falvey and Garreaud, 2009).

In the same area and a slightly further north (Callao, 12ºS/77ºW; Fig. 1), Sifeddine et al. (2008) also noted an increase (decrease) in the intensity of southerly winds during the CWP (LIA) attributed to a strengthened (weakened) SPSH. These

authors suggest that these conditions could be caused by changes in the ITCZ and the SPSH latitudinal position, being both displaced to the north (south) during the warm (cold) period. Changes in alongshore winds are also reported off Central Peru by Gutiérrez et al. (2009) and Briceño-Zuluaga et al. (2016), and in northern Chile (23ºS/70ºW; Fig. 1) by Flores-Aqueveque et al. (2015), who interpreted the recent increase in southerly wind intensity trend as a response to an expanded and/or intensified SPSH. According to Gutiérrez et al. (2009) and Briceño-Zuluaga et al. (2016), stronger winds could be related to

a northward displacement of the ITCZ-SPSH system during the CWP, and to an intensification of the Walker circulation associated to the current global warming. These authors also stated that during the LIA, the ITCZ was located southward of



its modern position, agreeing with the interpreted by Fleury et al. (2015) for the same area (11º-15ºS, Perú; Fig. 1). However, unlike the first authors, Fleury et al. (2015) noted a weakening in the Walker circulation since 1800 CE that suggests the influence of additional processes, probably related to a secular positive trend in tropical SST, accounting for this decrease.

Latitudinal changes in the ITCZ mean position, with a southward displacement during the LIA, have also been recorded in
other parts of the world as the Cariaco Basin (10ºN/ 66ºW; Fig. 1) in northern Venezuela (Haug et al., 2001; Peterson and Haug, 2006), Washington Island (4ºN/160ºW; Fig. 1), Palau Island (7ºN/134ºE; Fig. 1) and Galápagos Island (1ºS/89ºW; Fig. 1) in the Central Pacific (Sachs et al., 2009), Lake Malawi (10ºS/34ºE; Fig. 1) in eastern Africa (Johnson et al., 2001), and Abaco Island (26ºN/77ºW; Fig. 1) in northern Bahamas (van Hengstum et al., 2016).

Changes in the South American climate system are also recorded at higher latitudes. Using lacustrine sediments of Laguna
Aculeo (33ºS/71ºW; Fig. 1), Jenny et al. (2002) observed an increase in precipitation in Mediterranean Central Chile during the LIA, and interpreted it as an intensification of SWW. For the same period but using marine laminated sediments of Southern Chile (41ºS/74ºW; Fig. 1), Lamy et al. (2001) also noted an increment of the precipitation in southern Andes related to a northward displacement of the SWW, due to a weakened SPSH. The opposite occurs during a warm period, such as the Medieval Climate Anomaly (MCA).

Further south, Sepúlveda et al. (2009) reported a northward shift of the SWW belt during the LIA, according to a marine sedimentary record obtained from the Jacaf Fjord in northern Chilean Patagonia (44ºS/73ºW; Fig. 1). The same is interpreted from the analysis of a sediment core of Quitralco fjord (46ºS/73ºW; Fig. 1) by Bertrand et al. (2014), who found that during the LIA (1200-1500 CE), the SWW were gradually shifted northward and slightly moved poleward during the last decades, agreeing with recent trends in observed climatological data (e.g., Shindell and Schmidt, 2004). According to these authors,
late Holocene variations (latitudinal position and extent) in SWW were mostly driven by changes in the strength of the Polar Cell, which respond to changes in temperature at higher latitudes in the SH; this is in opposition to Lamy et al. (2001), who explained the changes in SWW based on variations in the tropical climate system (i.e., Hadley cell intensity).

Evidences of SWW variations are also observed in paleoclimate records at Subantarctic and Antarctic latitudes. The analyses of lacustrine sediments from Lago Guanaco, southwestern Patagonia (51ºS/73ºW; Fig. 1) developed by Moy et al. (2008)
and Moreno et al. (2009) indicate that precipitation increased during the LIA, related to more intense westerly winds. Moy et al. (2008) also suggest a poleward shift of the southern margin of the SWW belt synchronously with this intensification.

On the other hand, Schimpf et al., (2011) noted a decrease in precipitation during the LIA and interpreted it as a southward shift or a weakening in the core of the SWW, according to a stalagmite from the Southernmost Andes (53ºS/73ºW; Fig. 1). Recently, Browne et al. (2017), based on a fjord sediment core collected from the Auckland Islands (51ºS, 166ºE; Fig. 1),
interpreted a northward (southward) shift of the SWW during the LIA (MCA), suggesting that the synchronous changes in SWW observed on both sides of the Pacific could be controlled by atmospheric teleconnections between the low and high latitudes, modulated by the variability in AAO and ENSO. Finally, Koffman et al. (2014), based on dust particles found in an ice core from western Antarctica (79ºS; Fig. 1), interpreted that SWW are less (more) intense and/or displaced equatorward



(poleward) during the cold (warm) period, in response to both, surface temperature changes in the tropical Pacific and solar variability.

The apparent incongruity of the SWW changes observed among paleorecords could be partly explained by their locations. Then, the different proxies could be recording differential changes in the wind intensity between its edges and its core, 5 and/or latitudinal displacements of this belt.

## 3.2 Climate changes over southwestern South America since the Late Holocene according to GCMs

Figure 2 shows the difference between the meridional winds during the CWP and the LIA. In addition, the position of the ITCZ, the upper level STJ, the SWW at 850hPa, and the SPSH locations for the LIA (blue line), the CWP (magenta line) and the 21C (red line) are displayed.

10 The four-model ensemble annual and the austral summer and winter means show a strengthening of the SPSH during warm periods, expressed as an expansion of this high-pressure center, mainly to the west and south. This expansion is more evident when comparing CWP and 21C data, with its southern rim reaching higher latitudes as temperature increases. This behavior was already noted by Garreaud and Falvey (2009) using the PRECIS R2A regional climate model, and Kitoh et al. (2011) in high horizontal resolution (20 and 60 km grid) model projections. The opposite (i.e., a contracted SPSH) is observed during 15 the cold period (LIA), except for the austral winter (June-July-August; JJA), when SPSH for LIA and CWP are nearly identical.

With respect to the SPSH position, major differences can be observed in latitude between austral summer (December-January-February; DJF), when the maximum atmospheric pressure locates on average at ~33ºS, and austral winter (JJA) when it moves northward at ~27.5ºS (Table S1, Supplementary material). This behavior agrees with the seasonal variation 20 observed by Rahn and Garreaud (2013) using the Climate Forecast System Reanalysis (CFSR) data for the period 1979-2010, and by Barrett and Hameed (2017) using NCEP/NCAR reanalysis data for 1980-2013. The southward migration of the SPSH during austral summer would be attributable to a basin-wide increase in SLP around 40ºS induced by changes in the latitudinal position of the ITCZ and the presence of the South American monsoon (Rodwell and Hoskins, 2001).

Longitudinal changes are also seen at seasonal scale, with maximum values of SLP located at ~98ºW during austral summer 25 (DJF) and moving eastward in austral winter (JJA) at a ~89ºW on average (Table S1, Supplementary material). These values are close to those obtained by Rahn and Garreaud (2013) and Ancapichún and Garcés-Vargas (2015). The latter authors, using monthly NCEP/NCAR reanalysis data for the period 1949-2012, identified that over the past decade, the SPSH has intensified (at a rate of 1.36 hPa per decade) and shifted towards the southwest, with its maximum located at ~37ºS and ~108ºW during February and March, when the SPSH intensity is maximum. On the contrary, during May, when the SPSH 30 shows its weakest intensity, it locates further north (~26ºS) and closest to the continent (~86ºW). However, these observations are not in agreement with results from Barrett and Hameed (2017), who reported higher SLP values during austral spring (September to November), and an eastward migration during austral summer, whereas weaker values and a westward shift are observed in austral autumn (March to May). Since both works use NCEP/NCAR reanalysis, this



discrepancy could arise from the different methodologies used in each study to calculate SPSH position and/or the different periods studied.

Nevertheless, from our results it should be highlighted that, regardless of seasonality, as temperature increases, a poleward and westward shift of the maximum value of SLP is observed, following the expansion trend. This coincides with Gillett and

Fyfe (2013), Rykaczewski et al. (2015) and Lu et al. (2007), who indicated that SLP fields would tend to move poleward in response to increased greenhouse gas concentrations.

The SPSH expansion during the CWP period is accompanied with weaker meridional winds in front of northern and central Chilean coast at annual scale and during austral summer, where this difference is more remarkable. In austral winter, the opposite situation is developed, with an intensification of the coastal alongshore winds in northern to central Chile and a

reduction of the intensity south of ~35ºS. This effect could be explained by the fact that the SPSH expands to mid-latitudes and shows no significant changes at subtropical latitudes, producing an enhanced meridional pressure gradient that lead a strengthening of the low-level coastal jet (Garreaud and Falvey, 2009).

Wind intensity changes are also described by Belmadani et al. (2014), who analyzed the effect of global warming on the upwelling-favorable winds along the Peruvian and Chilean coast by using the Laboratoire de Météorologie Dynamique

(LMDz; Hourdin et al., 2006) global circulation model downscaled to a resolution of 0.5º x 0.5º. According to these authors, the increase in the alongshore winds from central to southern Chile is related to a poleward shift and/or an intensification of the maximum latitudinal pressure gradient associated to similar changes in the SPSH or the HC.

According to Garreaud and Falvey (2009), the strengthening of the alongshore wind speed will continue throughout the 21th century. This reinforcement is characterized by a marked seasonality in which the stronger meridional winds occur between

37º-41ºS during austral spring-summer, and migrate to subtropical latitudes in the austral fall-winter season, as result of an important increase in SLP (2-3 hPa) developed towards the southern rim of the SPSH. The authors also stated that by the end of this century, austral spring atmospheric coastal jet events will be more frequent and longer in duration that present-day events.

These results agree with the hypothesis of Rykaczewski et al. (2015). This hypothesis postulates that the anthropogenic

influence on seasonal changes and shifts of the geographic position of the major atmospheric high-pressure centers will affect the intensity of upwelling-favorable winds.

On the other hand, the upper level STJ, present at annual scale and during austral winter, displaces to the north as the temperature increases, which is more evident in the annual mean (Fig. 2). This behavior can be interpreted as a contraction of the tropics in response to anthropogenic forcing, and contradicts not only the general tendency of climate models of a

poleward shift under the current global warming (e.g., Meehl et al., 2007; Woollings and Blackburn, 2012), but also findings from recent observations. In fact, the analysis of total ozone measurements from satellite and sounding data discussed by Hudson (2012) estimates a poleward displacement of the STJ of 3.7º±0.3 latitude in NH, and 6.5º±0.2 latitude in SH, between 1979 and 2010.





Changes in SPSH also affect higher latitudes. As can be seen in Fig. 2, the expansion (contraction) of the SPSH observed during relative warmer (colder) periods coincides with a southward (northward) displacement of the core of SWW. In addition, the SWW are also stronger (weaker) during warmer (colder) periods (not shown). Therefore, an expanded SPSH acts to 'block' the SWW belt, shifting it southward. This behavior is observed at annual and seasonal mean scales, being
more pronounced during austral summer. This is in agreement with Garreaud et al. (2009), who used meteorological station and atmospheric reanalysis, and Swart et al. (2015), who considered thirty CMIP5 models and six reanalysis datasets. Garreaud et al. (2009) also pointed out that a contraction (expansion) of this wind belt occurs in parallel to southward (northward) seasonal shifts. However, since we only analyzed the position of its core, this feature cannot be observed from our results.

At longer timescales, Toggweiler et al. (2006) highlighted the general relationship between the position of the westerlies and global temperature. According to these authors, under warm conditions (e.g., the CWP), the SWW shift poleward while in cold climate periods (e.g., the Last Glacial Maximum), the westerlies moves equatorward, due to an increase of sea-ice around Antarctica (Bently et al., 2009).

The southward migration, together with the increase of its core strength, experienced by the SWW during the last decades
has been related to the shift to an increasingly positive phase of the SAM (e.g., Thompson et al., 2000; Thompson and Solomon, 2002; Marshall et al., 2004; Garreaud et al., 2009; Swart and Fyfe, 2012) as response to changes in stratospheric ozone and greenhouse forcing (Thompson and Solomon, 2002; Gillett and Thompson, 2003; Shindell and Schmidt, 2004; Ihara and Kushnir, 2009; Son et al., 2010; Gillett and Parker, 2013). This confirms what was previously stated by Pittock (1978) and Aceituno et al. (1993) who proposed that the position of the SWW depends on the location of the SPSH and the
circum-Antarctic low-pressure belt.

The trend in the SWW is also observed in several model projections for the 21th century, which highlight the tendency to a poleward shift and an intensification of the SWW related to the anthropogenic global warming (e.g., Fyfe and Saenko, 2006; Ihara and Kushnir, 2009; Kitoh et al., 2011; Chavaillaz et al., 2013; Bracegirdle et al., 2018).

Changes associated to SWW variations have already shown visible effects in southern SA, as an increase of desertification in
northern Chile (Salinas and Mendieta, 2013; Ortega et al., 2019) and a notable decrease of annual precipitation from central Chile to central Patagonia (Boisier et al., 2018), exhibiting important impacts on socio-economic activities in this region. Moreover, our results suggest that consequences related to SWW changes during the next decades, not only in southern SA (e.g., Vera et al., 2006; Meehl et al., 2007) but also at global scale, for example through the release of $CO_2$ from the Southern Ocean to the atmosphere (e.g., Menviel et al., 2018; Saunders et al., 2018), could be expected.

**3.3 Model simulations *versus* present-day atmospheric observations**

The analysis of the main present-day (1979-2009) climate features of southwestern SA from ERA-Interim reanalysis shows that some differences arise when comparing this dataset with model simulations for a comparable period (1970-2000). In general terms, climate models seem to underestimate the present day SPSH extension (Fig. S2, Supplementary material),





especially during austral summer (DJF) when reanalysis shows a SPSH more expanded than simulated conditions, being more similar to 21C projections. With respect to the location of the SPSH, it can be noted that, although latitudinal SPSH position is very similar in both datasets, major differences can be observed in its longitudinal position (Table S1, Supplementary material). Maximum values of the SLP in reanalysis data locate further west than those of the models at

annual as well as seasonal scale, being this difference more remarkable during austral summer, reaching around 11 degrees longitude.

The position of the STJ is another climate feature that differs between reanalysis and climate models. For reanalysis data, the latitudinal position of the STJ when it reaches the continent is around 27ºS during the austral winter and 29ºS in the annual mean, being between 3º and 6º latitude north, respectively, of those represented by historical simulations (i.e., ~30ºS during

austral winter and ~35ºS in the annual mean) (Fig. S2, Supplementary material). According to the results obtained by Hudson (2012), the SH STJ position varied between ~30º and ~45ºS between 1979 and 2010. This range is in good agreement with the values obtained from climate models, suggesting that ERA-Interim reanalysis dataset underestimates the STJ meridional position.

At higher latitudes, ERA-Interim reanalysis data differs from models in the representation of the present-day behavior of the

SWW (Fig. S2, Supplementary material), indicating a position further south of its core, when this wind belt meets the continent (~54ºS) during all year round, showing almost no seasonal differences. Similar results are obtained by Swart et al. (2015), who compared surface winds and satellite-based data, finding that several reanalysis overestimate recent observed trends. Moreover, these authors also found that long-term trends in SLP and winds observed in diverse reanalysis datasets for the SH, especially in the southeastern Pacific, limit its applicability as a tool for validating these features in model

simulations.

**3.4 The link between the ITCZ and SPSH changes**

Different paleoclimate studies have linked changes of the ITCZ-SPSH system to variations in the SPSH (Sifeddine et al., 2008, Gutiérrez et al., 2009, Briceño-Zuluaga et al., 2016, Fleury et al., 2015, Flores-Aqueveque et al., 2015). To evaluate the role of the ITCZ on SPSH changes, we analyze the position of the ITCZ during the three considered periods. Figure 2

shows that, in general, model simulations represent an ITCZ that locates north of the equator (10ºN) during austral winter (JJA), and moves southward (~5ºS) in austral summer (DJF). This range is very similar to the observed present-day conditions, in which ITCZ displaces seasonally from 9ºN to 2ºS (Denniston et al., 2016). However, according to Schneider et al. (2014), the ITCZ moves between 9ºN in austral winter (JJA) and 2ºN in austral summer (DJF) over the central Atlantic and Pacific oceans, suggesting that the considered models locates the ITCZ further south of its present-day position,

especially in austral summer. In fact, the bias of global models representing the ITCZ and its different causes has been widely discussed in literature (e.g., Hwang and Frierson, 2013; Hirota and Takayabu, 2013; Colas et al., 2012; Gordon et al., 2000; Ma et al., 1996).





In Fig. 2, it can also be observed that the ITCZ shifts southward over the Pacific as temperature increases, at annual and seasonal scale, being this difference more marked during austral summer. Over the Atlantic, the opposite occurs: the ITCZ migrates slightly to the north during the CWP, but it does not exhibit significant differences between the LIA and the CWP. Furthermore, we analyze the meridional mass stream function (as a proxy of the HC strength) and the zonal mean

precipitation during the LIA and the CWP (Fig. 3). Our results show a southern branch of the HC that intensifies during the cold period (LIA) and weakens in the current period at annual and seasonal scale (Fig. 3). In addition, no evident shifts of the northern and southern HC branches are observed between both periods.

A comparison of the intensity of precipitation rate between the LIA and the CWP indicates no significant differences between periods. However, as can be noted in both cases (Fig. 3), the maximum precipitation during the CWP is slightly

higher than in the LIA. In addition, zonal winds at 200 hPa show almost no differences between the LIA and the CWP, but appears to be located higher (lower) at poleward (equatorward) latitudes during the cold period.

These results contradict the mechanism proposed by Lee et al. (2011), which states that the ITCZ moves southward and the southern branch of the HC becomes weaker, decreasing the intensity of the STJ, as a response of NH (i.e., North Atlantic) cooling. These authors indicate that a colder NH increases the north-south surface pressure inter-hemispheric gradient,

driving an anomalous southward equatorial flow and shifting the annual mean position of the ITCZ southward. According to Sachs et al. (2009), this shift would have reached it southernmost position, of up to 500 km, during the LIA (1400-1850 AD). According to literature, the ITCZ has shown important changes during the LM. Regarding the ITCZ position, Schneider et al. (2014) stated that during the LIA, the ITCZ experienced a southward shift, explained by a cooling of the NH extratropics relative to its southern counterpart. However, this shift is not always supported by this mechanism. The same authors

indicate that ITCZ migrations are modulated by ENSO. In fact, during the transition from La Niña to El Niño, or during strong El Niño events, the NH extratropics warm around 0.08°C more than SH extratropics (Hansen et al., 2010), but contrary to what should be expected, the ITCZ moves southward. Schneider et al. (2014) justify this behavior considering variations in the tropical net energy input to the atmosphere. Slight changes in the atmospheric energy flux can produce substantial ITCZ migrations, being this sensitivity one of the possible explanations for the difficulty that current climate

models have in simulating the ITCZ position (Schneider et al., 2014).

On the other hand, Yan et al. (2015), based on paleohydrology records of western Pacific and climate models, proposed that instead of a meridional shift, a contraction of the tropical rain belt (i.e., the latitudinal range over which the ITCZ seasonally moves) occurs during the LIA. More generally, these authors proposed that over the western Pacific, the ITCZ expands and contracts over decadal to centennial timescales in response to external forcing, rather than showing a latitudinal migration.

The same is observed by the analysis of stalagmites of southern China discussed by Denniston et al. (2016), who pointed out that the latitudinal movement range of the ITCZ expanded and contracted during the last 3,000 years, in a process that operates at multidecadal to centennial scales.

Reanalysis data for the 20th century also supports the idea that the ITCZ does not migrate significantly with climate changes. In this sense, D'Agostino and Lionello (2016) indicated that since 1979, the ITCZ shows no significant trend. However, they



observed that the southern branch of HC migrates southward mainly during austral summer, while the northern edge shows no clear tendency. This would imply a SH STJ that shifts poleward as temperature increases, features that are not observed in Fig. 2 (which actually shows the opposite) or Fig. 3 (which shows no STJ migrations during the different periods).

The analysis of 21st century climate projections presented by Lu et al. (2007, 2008) and Seo et al. (2014) suggest a
weakening of the HC and its poleward expansion, associated with global warming. According to the range of 21st century simulations analyzed by D'Agostino et al. (2017), the HC widens and weakens with global warming, the changes in the tropical mean temperature being the best predictor of HC variations.

In summary, our analyses indicate that the climate models considered in this study do not adequately represent the ITCZ and HC changes reported in other studies. According to Schneider et al. (2014), biases in the model representation of important
features such as clouds could distort the atmospheric energy flux and account for the difficulties representing ITCZ changes. In particular, Fig. 3 shows that the four models considered in this work exhibit a double peak of maximum precipitation between 8ºN and 8ºS. This is observed at annual scale, when both peaks present the same amplitude (~6.7 mm/day), as well as in austral winter (JJA), when this behavior is more evident, with a main peak at 8ºN, reaching ~8.5 mm/day, and a secondary maximum of ~3.8 mm/day at 6ºS. Such bias is well-known as the "double ITCZ bias". Although some studies
suggest this bias has been reduced from CMIP3 to CMIP5 models (Hirota and Takayabu, 2013), it is still a systematic bias in current climate models (Hwang and Frierson, 2013). Among the different possible causes linked to this bias are a cloud deficit over the southern hemisphere (Hwang and Frierson, 2013), insufficient cooling by ocean mesoscale eddies from the upwelling region (Colas et al., 2012), warm SSTs biases in the coastal upwelling of Peru (Gordon et al., 2000), and an underestimation of stratocumulus clouds over Peru (Ma et al., 1996).

**4. Concluding remarks**

Our results suggest the occurrence of important changes in major climatic features of the western margin of South America during the Late Holocene. In particular, according to several paleoclimate proxies of southwestern SA and four CMIP5/PMIP3 models, we infer that important differences in SPSH strength and position, alongshore winds intensity and SWW position, occurred between two of the most remarkable global climate events of the LM: the LIA and the CWP.

The SPSH has changed from a contracted condition during the LIA period to an expanded state, with its southern rim reaching higher latitudes, during the CWP. Coinciding with these changes, coastal meridional winds appear to be stronger (weaker), south (north) of ~35º in the prevailing warm conditions (CWP). The opposite is observed during the cold period (LIA). The belt of SWW shows a poleward position in the CWP while it displays a northward shift during the LIA. These changes are in good agreement when comparing model results with proxy evidence from southwestern SA for the LIA and
the present-day climate. This is also ratified by paleoclimate reconstructions developed in other places of the SH and the tropical area of the NH. Figure 4 displays a scheme that summarizes these changes.



When evaluating historical experiments against ERA-Interim reanalysis data, important differences can be observed. Reanalysis data presents a SPSH more expanded, especially during austral summer, than that represented by climate models. Reanalysis also differs in representing the present-day behavior of the STJ, which is located some degrees north of those displayed by simulations and observations. In addition, reanalysis shows biases regarding the SWW core, which is located

further south when this wind belt meets the continent all year round.

The effects of global warming would continue impacting the South American climate. This is suggested by climate model projections, which indicate that SPSH would continue to expand throughout the 21st century. This expansion would also affect the intensity of the alongshore winds and the position and the strength of the SWW belt.

According to literature, all these changes seem to be related mainly to changes in the ITCZ position and the Hadley

circulation, but also to extratropical atmospheric dynamics. In fact, previous studies have shown that the ITCZ moved southward relative to its mean position during the LIA, as a response of NH cooling, and has been displaced northward (and/or expands its latitudinal range of movement) in the present-day warm conditions (Fig. 5). However, the models considered in this work fail in adequately representing these features, being not consistent with theory or the paleorecord evidence. In particular, model results show a weaker southern branch of the HC during the CWP in comparison to the LIA,

and show no significant changes in the Atlantic ITCZ during both periods. Biases in representing some features in models that distort the atmospheric energy flux could account for the difficulties exhibited by the considered climate models in representing correctly the ITCZ. However, these models suggest that the Pacific ITCZ shifts southward as global temperature increases and the STJ changes from a southward position during the LIA to a northward location during the CWP, and continues moving to the north by the end of the 21st century.

Finally, our results provide important insights about the regional impacts of the ongoing climate change in the SH, by integrating paleoclimate records of southwestern South America and global climate models. In view of the results from this work, we highlight the need to establish adequate mitigation and adaptation strategies to prevent the impacts of current and future climate change in southwestern South America.

**Author contribution**

V. Flores-Aqueveque and M. Rojas conceived of the central idea of this manuscript. M. Rojas, C. Aguirre and P.A. Arias developed the model code and performed the simulations. V. Flores-Aqueveque carried out the paleoclimate/paleoenvironmental record review. C. González contributed with the reanalysis data. V. Flores-Aqueveque prepared the manuscript and all authors discussed the results and contributed to the final work.



**Competing interest**

The authors declare that they have no conflict of interest.

**Data availability**

GCMs and Era-Interim reanalysis data used in this paper can be freely accessed on line at the following web addresses:
https://esgf-node.llnl.gov/projects/cmip5/ and https://www.ecmwf.int/en/forecasts/datasets/reanalysis-datasets/era-interim, respectively.

**Acknowledgements**

This work was partially funded by Millennium Science Initiative of the Ministry of Economy, Development and Tourism, grant "Nuclei for Paleoclimate". Valentina Flores-Aqueveque was supported by FONDECYT grant N°1191942 and grants
from Santander Universidades (Beca Iberoamérica) and Alianza del Pacífico. Maisa Rojas was supported by FONDECYT Nº1171773. Catalina Aguirre acknowledges support by CONICYT through PAI program N° 79150062 and FONDECYT grant N°11171163. Paola A. Arias was funded by grant PRG2017-16264 provided by Comité para el Desarrollo de la Investigación (CODI) at Universidad de Antioquia, Colombia.

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

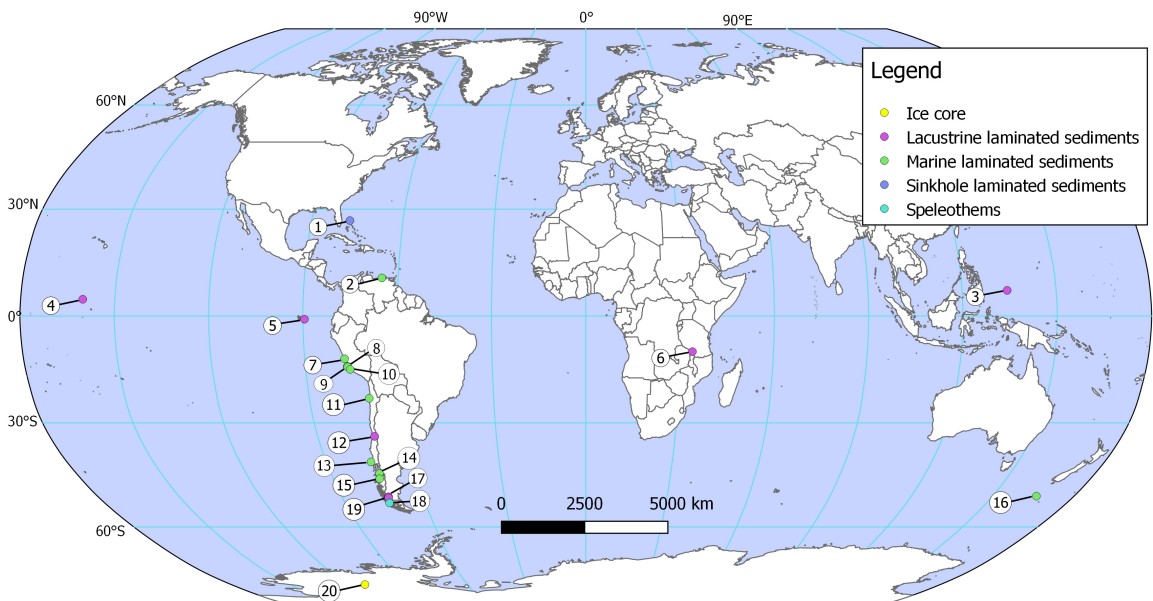

**Figure 1: Global distribution of the paleoclimate and paleo-environmental records used in this work. See Table 1 for number**
10  **identification.**





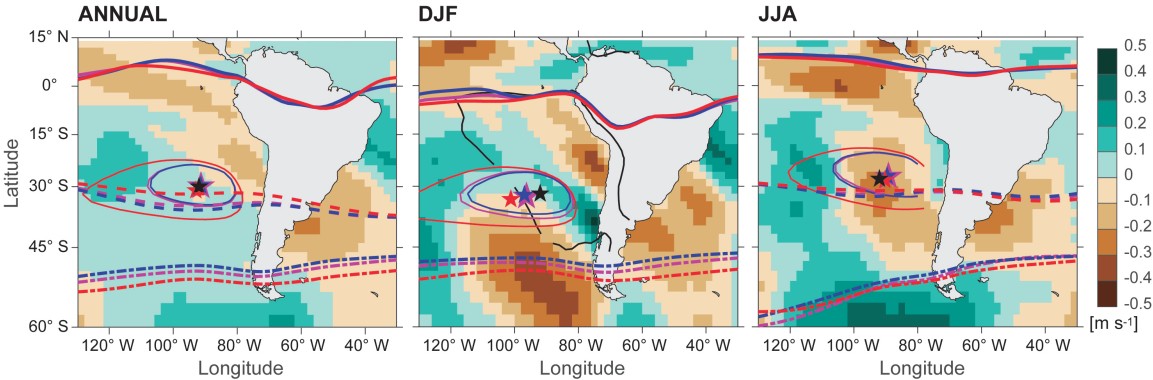

**Figure 2: Ensemble annual and seasonal means of southwestern South America climate system during the LIA (blue lines), the CWP (magenta lines) and 21C (red lines). Background colors show the meridional winds difference LIA minus CWP. DJF: austral summer, JJA: austral winter, ITCZ: Intertropical Convergence Zone (thick solid lines), SPSH: Subtropical South Pacific High (thin solid lines), STJ: Subtropical jet (dashed lines), SWW: Southern Westerly Winds (dash-dotted lines). Stars indicate the location of maximum SLP over the SPSH (i.e. SPSH location).**

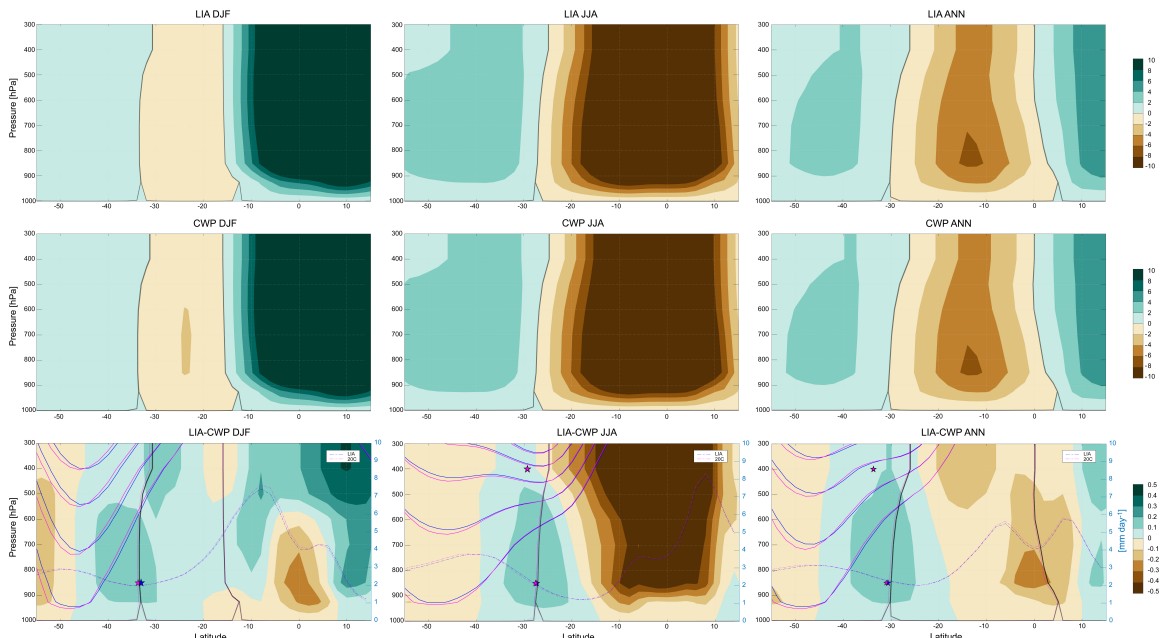

**Figure 3: Mass stream function ensemble mean during the LIA, CWP and LIA-CWP (shades), for austral summer (DJF), austral winter (JJA) and annual mean (ANN). Zonal mean precipitation (dash-dotted lines), zonal mean winds (solid lines), subtropical ridge position (lower stars) and latitude of maximum zonal winds at 200 hPa (upper stars) during the LIA (blue) and the CWP (magenta) are also shown.**





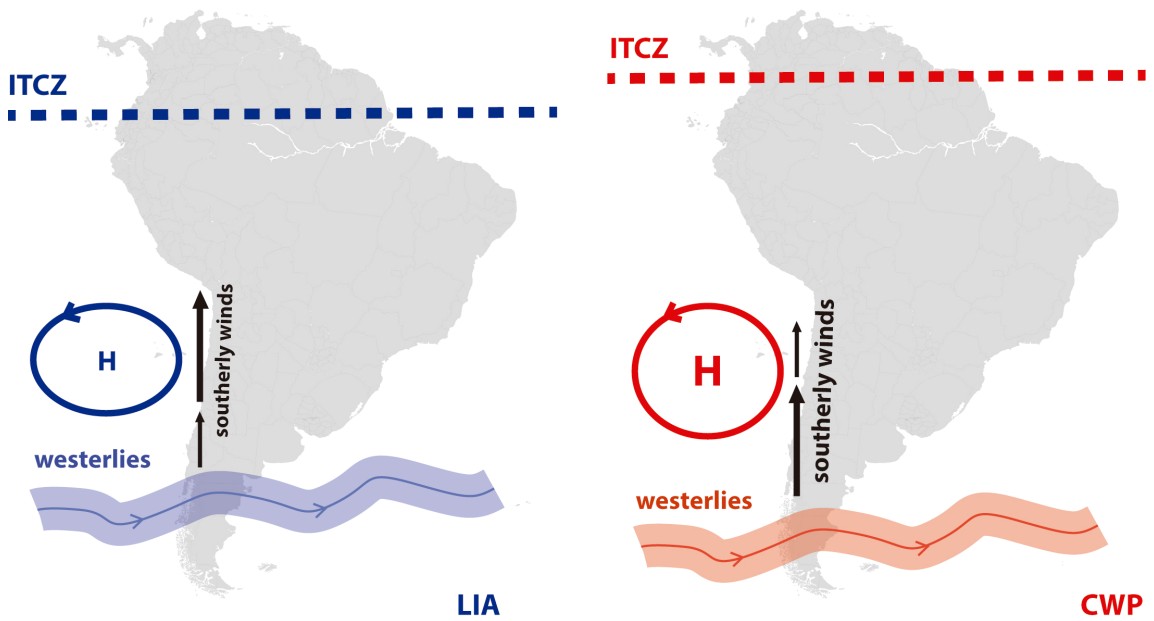

**Figure 4: Schematics of the changes proposed in this work. The SPSH has changed from a contracted condition during the LIA period to an expanded state, with its southern rim reaching higher latitudes, during the CWP. Coastal meridional winds appear to be stronger (weaker), south (north) of ~35º in the prevailing warm conditions during the CWP. The opposite is observed during**
5 **the cold period (LIA). The belt of SWW shows a poleward position in the CWP while it displays a northward shift during the LIA.**



| Model | Horizontal resolution | LIA period | Reference |
| --- | --- | --- | --- |
| bcc-csm-1 | 2.8ºx2.8º | 1590-1790 | Wu et al. (2013); Xin et al. (2013) |
| CCSM4 | 0.9ºx1.25º | 1710-1810 | Gent et al. (2001) |
| IPSL-CM5A-LR | 1.9ºx3.75º | 1630-1710 | Dufresne et al. (2013) |
| MRI-CGCM3 | 1.1ºx1.12º | 1510-1620 | Yukimoto et al. (2011) |

**Table 1: Characteristics and definition of the LIA period for each LM model simulation used. Modified from Rojas et al. (2016).**





| Nº | Record type | Location | Lat (°) | Lon (°) | Spanned period | Temporal resolution | Reference |
|---|---|---|---|---|---|---|---|
| 1 | Sinkhole laminated sediments | Southeastern North America | 26.79 | -77.42 | last 3000 yrs | Multi-decadal | van Hengstum et al., 2016 |
| 2 | Marine laminated sediments | Northern South America | 10.7 | -65.2 | last ~14000 yrs | Inter-annual | Haug et al., 2001; Peterson and Haug, 2006 |
| 3 | Lacustrine laminated sediments | Central western Pacific Ocean | 7.2 | 134.4 | last ~500 yrs | Inter-decadal to centennial | Sachs et al., 2009 |
| 4 | Lacustrine laminated sediments | Central Pacific Ocean | 4.7 | -160.2 | last 3200 yrs | Inter-decadal to centennial | Sachs et al., 2009 |
| 5 | Lacustrine laminated sediments | Central eastern Pacific Ocean | -0.9 | -89.5 | last 1200 yrs | Inter-decadal to centennial | Sachs et al., 2009 |
| 6 | Lacustrine laminated sediments | Eastern Africa | -10.0 | 34.1 | last 700 yrs | Decadal | Johnson et al., 2001 |
| 7 | Marine laminated sediments | Southwestern South America | -12.0 | -77.2 | last 700 yrs | Sub-decadal to inter-decadal | Siffedine et al., 2008; Gutiérrez et al., 2009; Briceño-Zuluaga et al., 2016 |
| 8 | Marine laminated sediments | Southwestern South America | -14.1 | -76.5 | last 700 yrs | Sub-decadal to inter-decadal | Siffedine et al., 2008; Gutiérrez et al., 2009 |
| 9 | Marine laminated sediments | Southwestern South America | -14.3 | -76.4 | last ~2000 yrs | Inter-annual to decadal | Salvatecci et al., 2014; Briceño-Zuluaga et al., 2016 |
| 10 | Marine laminated sediments | Southwestern South America | -15.0 | -75.7 | last 1000 yrs | Inter-annual to multi-decadal | Fleury et al., 2015 |
| 11 | Marine laminated sediments | Southwestern South America | -23.1 | -70.5 | last ~250 yrs | Annual | Flores-Aqueveque et al., 2015 |
| 12 | Lacustrine laminated sediments | Southwestern South America | -33.8 | -70.9 | last 2000 yrs | Decadal to centennial | Jenny et al., 2002 |
| 13 | Marine laminated sediments | Southwestern South America | -41.0 | -74.5 | last ~7700 | Sub-decadal to inter-decadal | Lamy et al., 2001 |
| 14 | Marine laminated sediments | Southwestern South America | -44.3 | -72.9 | last 1750 yrs | Decadal | Sepúlveda et al., 2009 |
| 15 | Marine laminated sediments | Southwestern South America | -45.8 | -73.5 | last 1400 yrs | Multi-decadal to multi-centennial | Bertrand et al., 2014 |
| 16 | Marine laminated sediments | Southwestern Pacific | -50.8 | 166.1 | last ~5000 yrs | Centennial to millennial | Browne et al., 2017 |
| 17 | Lacustrine laminated sediments | Southwestern South America | -51.0 | -72.8 | last 5000 yrs | Decadal to multi-centennial | Moy et al., 2008; Moreno et al., 2009 |
| 18 | Lacustrine laminated sediments | Southwestern South America | -51.3 | -72.9 | last 3000 yrs | Centennial | Moreno et al., 2014 |
| 19 | Speleothems | Southwestern South America | -52.8 | -73.4 | last 5000 yrs | Sub-annual | Schimpf et al., 2011 |
| 20 | Ice core | Western Antarctica | -79.5 | -112.1 | last 2400 yrs | Sub-annual | Koffman et al., 2014 |

**Table 2: List of paleoclimate and paleoenvironmental records from southern South America and other regions used in this work, and their characteristics.**





| Period | Latitude | ITCZ | SPSH | Meridional winds | SWW | Reference |
|---|---|---|---|---|---|---|
| | 26.8 | Shifted southward | - | - | - | van Hengstum et al., 2016 |
| | 10.7 | Shifted southward | - | - | - | Haug et al., 2001; Peterson and Haug, 2005 |
| | 7.2 | Shifted southward | - | - | - | Sachs et al., 2009 |
| | 4.7 | Shifted southward | - | - | - | Sachs et al., 2009 |
| | -0.9 | Shifted southward | - | - | - | Sachs et al., 2009 |
| | -10.0 | Shifted southward | - | - | - | Johnson et al., 2001 |
| | -12.0 | Shifted southward | Shifted southward / Weakened | Weakened | - | Siffedine et al., 2008 |
| | -12.0 | Shifted southward | Weakened | - | - | Gutiérrez et al., 2009 |
| | -14.1 | Shifted southward | Weakened | Weakened | - | Briceño-Zuluaga et al., 2016 |
| | -14.3 | Shifted southward | Contracted | Weakened | - | Salvatecci et al., 2014; |
| Little Ice Age | -15.0 | Shifted southward | - | - | - | Fleury et al., 2015 |
| | -33.8 | - | Weakened | - | Intensified and/or shifted northward | Jenny et al., 2002 |
| | -41.0 | - | Weakened | - | Intensified and shifted northward | Lamy et al., 2001 |
| | -44.3 | - | - | - | Shifted northward | Sepúlveda et al., 2009 |
| | -45.8 | - | - | - | Shifted northward | Bertrand et al., 2014 |
| | -50.8 | - | - | - | Shifted northward | Browne et al., 2017 |
| | -51.0 | | | | Intensified and (southern edge) shifted southward | Moy et al., 2008 |
| | -51.0 | - | - | - | Intensified | Moreno et al., 2009 |
| | -52.8 | - | - | - | Weakened | Schimpf et al., 2011 |
| | -79.5 | Shifted southward | - | - | Weakened and/or shifted northward | Koffman et al., 2014 |
| | - | **Shifted southward** | **Weakened/contracted** | **Weakened** | **Shifted northward** | - |
| | 26.8 | Shifted northward | - | - | - | van Hengstum et al., 2016 |
| | -12 | - | - | Strengthened | - | Siffedine et al., 2008 |
| | -12 | Shifted northward | Shifted northward / strengthened | Strengthened | - | Gutiérrez et al., 2009 |
| | -14.1 | Shifted northward | Shifted northward / strengthened | Strengthened | - | Briceño-Zuluaga et al., 2016 |
| Current Warm Period (1970-2000) | -14.3 | Shifted northward | Stregthened | Strengthened | - | Salvatecci et al., 2014; |
| | -15 | Shifted northward | - | - | - | Fleury et al., 2015 |
| | -23.1 | - | Strengthened or expanded | Strengthened | - | Flores-Aqueveque et al., 2015 |
| | -45.8 | - | - | - | Shifted southward | Bertrand et al., 2014 |
| | -51.3 | - | - | - | Intensified and shifted southward | Moreno et al., 2014 |
| | -79.5 | - | - | - | Intensified and shifted southward | Koffman et al., 2014 |
| | - | **Shifted northward** | **Stregthened/expanded** | **Strengthened** | **Shifted southward** | - |

**Table 3: Climate reconstructions of southwestern South America for the LIA and the CWP based on paleoclimate records.**

