# Peer review of "South Pacific Subtropical High from the late Holocene to the end of the 21st century: insights from climate proxies and general circulation models"

_Climate of the Past, 2019_

## Referee Comment (RC1) · Anonymous Referee #1 · 12 Aug 2019

This study investigates changes in the South Pacific subtropical high (SPSH) using paleoclimate records, climate model simulations and ERA-Interim reanalysis data. The study is generally well designed and the methods and results are clearly described. The results are of relevance for understanding drivers of Southern Hemisphere circulation change, and evaluating model responses in past and future climates. I recommend publication subject to minor revisions as outlined below.

General Comments:

[Figure]

1. I am concerned about the use of such a small sample of models (4 models) to draw conclusions about changes in the SPSH. I cannot see why the authors could not use at least 6-8 models that have Last Millennium simulations, even excluding non-CMIP5 models HadCM3 and CSIRO Mk3L.

2. When comparing models for LIA, CWP and RCP8.5, it may not be informative to consider only the multi-model mean. If there is large model disagreement, the model mean change does not represent the changes of each model. Instead, calculating changes in a given variable for each model and then comparing these, e.g. using a scatter plot or box and whisker plot, may be more informative. The model spread also provides a measure of uncertainty. (See also specific comment for page 13 below).

3. The model evaluation compared with observations or reanalysis should be included earlier in the paper as it provides the justification for using the models to examine past and future climate. That is, swap the order of section 3.2 and 3.3. Then include a few sentences at the end of the model evaluation section about the strengths and weaknesses of models (also add a figure comparing observations and model climatologies).

Specific Comments:

Page 3, Section 1.1: It would be helpful to include a Figure or schematic showing the regional climatological circulation in austral summer and winter.

Page 3, line 12: "exceeding 45S" – does this mean extending poleward of 45S? It is not clear.

Page 4, line 25-28: There is reasonable evidence of a period of synchronous cooling between Northern and Southern Hemispheres, although this does not imply that the signal is synchronous on a regional scale. For example, Neukom et al. (2014) state that "simultaneous cold anomalies in both hemispheres are identified between 1571 and 1722". Perhaps provide some qualification here, or explain the difference between Southern Hemisphere and South American scale responses.

Page 5, line 19: Why did you only use 4 CMIP5/PMIP3 models when there are many more (8+) models available with the required simulations? You should comment on the limitation of relying on such a small number of models.

Page 6, line 21: You could also cite the new PAGES2K study here (PAGES2K Consortium, Nature Geosciences, 2019).

Page 8, line 12: I am not sure what is meant by "increment"

Page 10, line 5: How can SLP fields move poleward?

Page 11, line 33: Do you mean the 4 models evaluated in this study, or a larger sample of CMIP models?

Page 12, line 18: "long-term trends": do you mean spurious (incorrect) long term trends, or actual anthropogenic climate change trends?

Page 12, line 27: Denniston et al. (2016) is a study of the ITCZ in the Indo-Pacific region, so the position of the ITCZ in that study is not directly relevant to the ITCZ over South America.

Page 13, line 8 onwards: the lack of signal in the LIA and CWP comparison based on models may be due to model disagreement. If you are comparing the multi-model mean values only, you may be smoothing out changes in individual models. An alternative way to show the changes might be a scatter plot or box and whisker plot of changes for each individual model (for example, change in location of ITCZ or Hadley Cell edge versus area average change in precipitation). This would be even more informative if more than 4 models were used.

Page 15, line 1-5: How do the model biases impact on the simulated changes in past and future climate? Do they reduce confidence in the results?

Page 15, line 23: You could also comment on the need to improve model performance in the simulation of Southern Hemisphere circulation to provide more robust projections.

Figure 2: I found it difficult to distinguish the red and magenta lines. Perhaps use different colors?

Figure 3: These plots are quite small with very small labels and legends. It is also hard to see the overplotted contour lines. I suggest plotting the zonal mean precipitation and winds in a separate set of plots to make it easier to see (there should be space for more figures as the paper currently only has 4 figures).

Figure 4: In this study, you do not find a shift in the Atlantic ITCZ and find a southward shift in the Pacific ITCZ as temperatures increase (according to line page 15, lines 15-20) so I am not sure why the ITCZ is plotted south in both sectors during LIA compared with CWP?

Supplementary Figure 1: The green and magenta lines appear to be in the same location?

Supplementary Figure 2: The ERA-Interim climatology and the model climatology should be given in the main paper as this is an important part of evaluating model skill. Also, what are the stars? Also, the austral summer and winter lines appear to be swapped or wrongly labelled.

Technical Corrections:

Page 2, line 9: replace "interplays" with "interplay"

Page 2, line 16: replace "During last decades" with "During recent decades"

Page 2, line 25: delete "derived"

Page 3, line 6: replace "these evidences" with "this evidence"

Page 3, line 25: replace "to higher probability" with "with higher probability"

Page 5, line 2: replace "associated to" with "associated with" (and elsewhere)

Page 5, line 5: at the end of this line, I think "LM" is meant to be "LIA"?

Page 5, line 10: replace "several" with "numerous" or "many"

Page 5, line 12: replace "uniform period..." with "period of uniformly positive temperature trends" for clarity.

Page 6, line 6: replace "spanning time" with "time period"

Page 13, line 24: this sentence is unclear.

Page 15, line 12: figure 5 is actually figure 4.

References: Please indent or add space between references to separate.

References:

PAGES2K Consortium (2019), Consistent multidecadal variability in global temperature reconstructions and simulations over the Common Era, Nature Geosciences, 12, 643–649.

Please also note the supplement to this comment:
https://www.clim-past-discuss.net/cp-2019-69/cp-2019-69-RC1-supplement.pdf

---

## Referee Comment (RC2) · Anonymous Referee #2 · 20 Sep 2019

The study presents an analysis of the South Pacific Subtropical High (SPSH) in two periods of extreme climate in the Last Millennium, i.e. the Little Ice Age (LIA) and the present-day climate, referred to as Current Warm Period (CWP), and a future RCP8.5 scenario. The model analysis and paleoclimate records suggest that, in the CWP compared to the LIA, the SPSH expands poleward and the Southern Westerly Winds (SWW) along the coast off South America intensify and shift southward.

The manuscript presents a good literature review on the topic, the methods are sound, but I found the results are half supported by the analysis. Despite that, I believe the

manuscript contains important information for the region and I recommend publication after revisions. A few comments below.

Based on previous literature, the authors discuss the link between the ITCZ and the SPSH and hypothesize that the differences in the SPSH strength, position and associated SWW between LIA and CWP is related to changes in the ITCZ and Hadley circulation. However, climate models do not simulate the expected changes in the position of the ITCZ and ERA-Interim differs substantially from models. Therefore, the mechanisms proposed here cannot be verified. Even though changes in the ITCZ cannot be supported, Figure 4 depicts a nice schematic with different position in the ITCZ. At least a cautionary note should be added here to not mislead the readers.

As mentioned in the discussions, extratropical dynamics may also play a role in those changes. For example, could different sea-ice extension between the two periods a possible driver of the SPSH changes? In addition to that, the differential heating between land and ocean (see Fig. S1 for the difference in temperature between the South Pacific and South America) can create a pressure gradient that by geostrophy can accelerate the SWW. Exploring these other mechanisms could help in the interpretation of the paleoclimate records. Or at least it is worth a discussion in the manuscript.

ERA-Interim is used in this study. Note that this reanalysis has been superseded by the newest generation of ECMWF reanalysis, ERA5, and discontinued this year. Not a big deal for this study and the type of analysis presented here, but keep in mind when justifying dataset choice in future studies.

On page 10, p.30, this text needs rewording: "This behaviour can be interpreted as a contraction of the tropics in response to anthropogenic forcing, . . ." It is not intuitive that increase in greenhouse gases would lead to a contraction of the tropics. There is also no mechanism to explain how this can happen. In fact, climate models suggest the opposite, i.e. an expansion of the tropics due to increase in greenhouse gases. Rewording this sentence may be needed.

On page 15, para 10, there is a citation to Fig. 5 that is not in the manuscript.

Figure 2: The magenta and red lines are hard to distinguish. Figure 3: All axis labels need to be enlarged. Lines are too thin, barely can see the dashed lines.

---

## Author Comment (AC1) · 19 Oct 2019

Answer to Referee #2

1.- Based on previous literature, the authors discuss the link between the ITCZ and the SPSH and hypothesize that the differences in the SPSH strength, position and associated SWW between LIA and CWP is related to changes in the ITCZ and Hadley circulation. However, climate models do not simulate the expected changes in the position of the ITCZ and ERA-Interim differs substantially from models. Therefore, the

mechanisms proposed here cannot be verified. Even though changes in the ITCZ cannot be supported, Figure 4 depicts a nice schematic with different position in the ITCZ. At least a cautionary note should be added here to not mislead the readers. Figure 4 (now Figure 6) represent the integration of paleorecord information and models results for the extratropical climate features as contains elements of both sources of information. The misrepresentation of ITCZ in the models used in this work (and other models not analyzed here) is a known weakness of simulation models. However, reconstructions based on paleoclimate records are very consistent among them (all ITCZ reconstructions suggest that ITCZ was located further south during the LIA (n=12) and shifted equatorward in the CWP (n=5)) and in agreement with the expected physical mechanisms (e.g., Sachs et al., 2009; Lee et al., 2011; Schneider et al., 2014). In order to clarify the integrative character of Figure 6, a note in 'Concluding Remarks' was included. In addition, few sentences explaining the incongruencies observed for the tropical system between our models and recent observations and therefore, the need of complementary analyses, were also added in this section.

2.- As mentioned in the discussions, extratropical dynamics may also play a role in those changes. For example, could different sea-ice extension between the two periods a possible driver of the SPSH changes? In addition to that, the differential heating between land and ocean (see Fig. S1 for the difference in temperature between the South Pacific and South America) can create a pressure gradient that by geostrophy can accelerate the SWW. Exploring these other mechanisms could help in the interpretation of the paleoclimate records. Or at least it is worth a discussion in the manuscript. R: In fact, any factor (e.g., SAM seasonality, ENSO, changes of ice cover in Antarctica) that affects surface temperature can influence the general convection pattern, which controls position and extension of SPSH. In particular, as we discussed in section 3.3, according to Bently et al. (2009) the increase of sea-ice around Antarctica during cold periods would produce an equatorward shift of the SWW. This migration could block the SPSH limiting its expansion and latitudinal movement. The same 'blockling' effect could be produced by an increase of SWW intensity driven by a decrease in the latitudinal temperature gradient. We added a few sentences developing this idea in Results and Discussions (section 3.3).

3.- ERA-Interim is used in this study. Note that this reanalysis has been superseded by the newest generation of ECMWF reanalysis, ERA5, and discontinued this year. Not a big deal for this study and the type of analysis presented here, but keep in mind when justifying dataset choice in future studies. R: Although recently replaced, ERA-Interim is a widely used reanalysis which favors the comparison between our results and other works based on this same data set. Furthermore, we considered Era-Interim due to its spatial scale of 0.75°x0.75° which gives a better comparison with the interpolated horizonal scale (1°x1°) that we used in the models.

On page 10, p.30, this text needs rewording: "This behaviour can be interpreted as a contraction of the tropics in response to anthropogenic forcing, : : :" It is not intuitive that increase in greenhouse gases would lead to a contraction of the tropics. There is also no mechanism to explain how this can happen. In fact, climate models suggest the opposite, i.e. an expansion of the tropics due to increase in greenhouse gases. Rewording this sentence may be needed. R: This sentence was simplified and reworded.

On page 15, para 10, there is a citation to Fig. 5 that is not in the manuscript. R: Yes, it was a mistake. In this revised version we added a new figure at the beginning and the previous Figure 4 is now Figure 6.

Figure 2: The magenta and red lines are hard to distinguish. R: We changed magenta for a brighter color and we darkened the color red to increase contrast. In addition, color blue was replaced by cyan.

Figure 3: All axis labels need to be enlarged. Lines are too thin, barely can see the dashed lines. R: We enlarge labels and legends, and we also increase the thickness of all lines.

Please also note the supplement to this comment:
https://www.clim-past-discuss.net/cp-2019-69/cp-2019-69-AC1-supplement.pdf

[revised manuscript text omitted]

---

## Author Comment (AC2) · 19 Oct 2019

Answer to Referee #1

1. I am concerned about the use of such a small sample of models (4 models) to draw conclusions about changes in the SPSH. I cannot see why the authors could not use at least 6-8 models that have Last Millennium simulations, even excluding non-CMIP5 models HadCM3 and CSIRO Mk3L. R: We started with seven CMIP5/PMIP3 models but in order to carry out an adequate and rigorous comparison we had to discard the

other three due to the impossibility to compare the first ensemble member (r1i1p1) of models for which the (1) Last Millennium simulation, (2) the historical simulation, and (3) the RCP8.5 scenario, were available in open access mode. Even though we understand four model could be a small sample, these four are the only models that fit the criteria described in the methodology, in particular the open access principle. We are aware of the uncertainties generated, but our results show consistencies in the responses to the forcing. This fact can be observed in the new Figure 3 (previously Figure 2), in which we include the results for each model in addition to the model ensemble mean.

2. When comparing models for LIA, CWP and RCP8.5, it may not be informative to consider only the multi-model mean. If there is large model disagreement, the model mean change does not represent the changes of each model. Instead, calculating changes in a given variable for each model and then comparing these, e.g. using a scatter plot or box and whisker plot, may be more informative. The model spread also provides a measure of uncertainty. (See also specific comment for page 13 below). R: A boxplot and whisker plot would not be statistically meaningful therefore not the best way to show models dispersion in consistency with our sample. As boxplot provides a five-number summary, n=4 models do not meet the minimum requirements for its application. In fact, the minimum recommended sample size for boxplot is n>20. Although, we agree that the multi-model mean may not very informative, consequently for a better visualization of the results, we modify the previous Figure 2 (now, Figure 3) to include the results for each model in addition to multi-model mean.

3. The model evaluation compared with observations or reanalysis should be included earlier in the paper as it provides the justification for using the models to examine past and future climate. That is, swap the order of section 3.2 and 3.3. Then include a few sentences at the end of the model evaluation section about the strengths and weaknesses of models (also add a figure comparing observations and model climatologies). R: We rearrange the order of both sections and included a new paragraph of synthesis

about the comparison of our results and previous studies. We also added a new figure (Figure 1) including observations and models climatology.

Specific Comments

Page 3, Section 1.1: It would be helpful to include a Figure or schematic showing the regional climatological circulation in austral summer and winter. A new figure (Figure 1) showing the ERA-Interim climatology for the period 1979-2009, highlighting the main climatic features (SLP, winds) was added.

Page 3, line 12: "exceeding 45S" – does this mean extending poleward of 45S? It is not clear. The phrase "exceeding 45°S" was replaced by "extending poleward of 45°S"

Page 4, line 25-28: There is reasonable evidence of a period of synchronous cooling between Northern and Southern Hemispheres, although this does not imply that the signal is synchronous on a regional scale. For example, Neukom et al. (2014) state that "simultaneous cold anomalies in both hemispheres are identified between 1571 and 1722". Perhaps provide some qualification here, or explain the difference between Southern Hemisphere and South American scale responses. R: A text about the work of Neukom et al. (2014) was included in section 1.2. In addition text was modified highlighting the difference between hemispheric and regional scales.

Page 5, line 19: Why did you only use 4 CMIP5/PMIP3 models when there are many more (8+) models available with the required simulations? You should comment on the limitation of relying on such a small number of models. R: To answer the first part of this question please referred above in the answer for comment 1. To answer the second part, a few sentences in 'Concluding remarks' about the need of future complementary analyses for reducing the uncertainties were included.

Page 6, line 21: You could also cite the new PAGES2K study here (PAGES2K Consortium, Nature Geosciences, 2019). Thanks for the recommendation. The new reference was added.

Page 8, line 12: I am not sure what is meant by "increment" The word "increment" was changed by "increase"

Page 11, line 5: How can SLP fields move poleward? The sentence was replaced by "higher values of SLP would tend to move poleward"

Page 11, line 33: Do you mean the 4 models evaluated in this study, or a larger sample of CMIP models? The sentence was clarified by adding the phrase "the four models analyzed here"

Page 12, line 18: "long-term trends": do you mean spurious (incorrect) long term trends, or actual anthropogenic climate change trends? R: Swart et al. (2015) found spurious long-term trends in all (6) reanalysis they studied. The term "spurious" was included before "long-term trends" to clarify.

Page 12, line 27: Denniston et al. (2016) is a study of the ITCZ in the Indo-Pacific region, so the position of the ITCZ in that study is not directly relevant to the ITCZ over South America. R: The work of Denniston et al. (2016) was cited here only for characterizing the latitudinal range on movement of the ITCZ in a general way. In addition, we also consider it to complement the observations of Yan et al. (2015) as both authors point out a change in the latitudinal range (contraction-expansion) of the tropical rain belt. While Yan et al. (2015) takes the western Pacific region and Denniston et al. (2015) the Indo-Pacific region, the feature con be interpreted independently of its longitudinal location because the consequence of this effect must be considered to understand the influence of the tropical dynamics at higher latitudes.

Page 13, line 8 onwards: the lack of signal in the LIA and CWP comparison based on models may be due to model disagreement. If you are comparing the multi-model mean values only, you may be smoothing out changes in individual models. An alternative way to show the changes might be a scatter plot or box and whisker plot of changes for each individual model (for example, change in location of ITCZ or Hadley Cell edge versus area average change in precipitation). This would be even more informative if

more than 4 models were used. R: As we explained in the answer to Comment 2, a sample size of n<20 is not recommended for boxplot and whisker plot. For this reason and to improve the model disagreement comparison, we include as supplementary figure (Supplementary figure 2) the difference of mass stream function between LIA and CWP (LIA-CWP) for the annual mean, DJF and JJA for each individual model. This figure also displays the zonal mean precipitation, zonal mean winds, subtropical ridge position and latitudinal position of maximum zonal winds at 200 hPa.

Page 15, line 1-5: How do the model biases impact on the simulated changes in past and future climate? Do they reduce confidence in the results? R: When analyzing past and future changes with respect to historical simulations, we would be considering the same biases and our main point of interest is how different aspects of circulation vary within the same model. If there are systematic biases in the models, these would be reproduced in historical simulations -which we compared with reanalysis- as well as in past simulations (last millennium) and future projections. Moreover, the aim of this work is to establish a relative comparison of SA climate under a cold period (LIA), a warm period (CWP), and warmer conditions (RCP8.5) rather than a quantification of these changes. We include a few sentences in 'Concluding Remarks' developing this idea.

Page 15, line 23: You could also comment on the need to improve model performance in the simulation of Southern Hemisphere circulation to provide more robust projections. R: Thanks for the advice. We included a few sentences about that in Concluding Remarks.

Figure 2: I found it difficult to distinguish the red and magenta lines. Perhaps use different colors? R: We changed magenta for a brighter color (fuchsia) and we darkened the color red to increase contrast between both. In addition, color blue was replaced by cyan.

Figure 3: These plots are quite small with very small labels and legends. It is also hard

to see the overplotted contour lines. I suggest plotting the zonal mean precipitation and winds in a separate set of plots to make it easier to see (there should be space for more figures as the paper currently only has 4 figures). R: Due to the close relation between the Hadley Cell, the zonal mean winds and the zonal mean precipitation we determined that these variables must be presented in the same plot. However, to facilitate image viewing we change the color blue (LIA) by cyan which is more contrasting and we enlarge labels and legends.

Figure 4: In this study, you do not find a shift in the Atlantic ITCZ and find a southward shift in the Pacific ITCZ as temperatures increase (according to line page 15, lines 15-20) so I am not sure why the ITCZ is plotted south in both sectors during LIA compared with CWP? R: Figure 6 (formerly Figure 4) shows a scheme resulting from the integration of both, paleoclimate records and model results. As can be read on section 3.4 models fails in representing tropical dynamics which is a known weakness of simulation models. However, reconstructions based on paleoclimate records are consistent among them and in agreement with the expected physical mechanisms exposed in literature (e.g., Sachs et al., 2009; Lee et al., 2011; Schneider et al., 2014). For this reason, Figure 6 displays an ITCZ located in a northern/southern position during the CWP/LIA.

Supplementary Figure 1: The green and magenta lines appear to be in the same location? R: Unfortunately, the scale prevents to appreciate better the small variations provided for the models in the position of the ITCZ during the LIA and the CWP in the figure 1S. However, as the display of the inter-hemispheric temperature differences is priority, we included a global scale map.

Supplementary Figure 2: The ERA-Interim climatology and the model climatology should be given in the main paper as this is an important part of evaluating model skill. Also, what are the stars? Also, the austral summer and winter lines appear to be swapped or wrongly labelled. R: We added the model climatology and we include it in the main paper (from now Figure 1). In figure caption the meaning of stars (= maximum

value of slp) was added, and the labels of summer (red) and winter (blue) lines were corrected.

Technical Corrections Page 2, line 9: replace "interplays" with "interplay" R: The word "interplays" was replaced.

Page 2, line 16: replace "During last decades" with "During recent decades" R: The word "last" was replaced by "recent".

Page 2, line 25: delete "derived" R: The word "derived" was deleted.

Page 3, line 6: replace "these evidences" with "this evidence" R: The phrase "these evidences" was replaced by "this evidence".

Page 3, line 25: replace "to higher probability" with "with higher probability" R: The word "to" was changed by "with".

Page 5, line 2: replace "associated to" with "associated with" (and elsewhere) R: The phrase "associated to" was replaced by "associated with" throughout the entire text.

Page 5, line 5: at the end of this line, I think "LM" is meant to be "LIA"? R: Yes, thank you. The right initials are "LIA".

Page 5, line 10: replace "several" with "numerous" or "many" R: "Several" was changed by "many".

Page 5, line 12: replace "uniform period: : :" with "period of uniformly positive temperature trends" for clarity. R: The sentence was replaced.

Page 6, line 6: replace "spanning time" with "time period" R: "Spanning time" was changed by "time period"

Page 13, line 24: this sentence is unclear. R: The sentence was changed to: "On the other hand, Yan et al. (2015), based on paleohydrology records of western Pacific and climate models, proposed that during the LIA a contraction of the tropical rain belt

(i.e., the latitudinal range over which the ITCZ seasonally moves) occurs, instead of a meridional shift as has been described."

Page 15, line 12: figure 5 is actually figure 4. R: Yes, in fact is figure 4. This error was corrected.

References: Please indent or add space between references to separate. R: Indent was added to references.

Please also note the supplement to this comment:
https://www.clim-past-discuss.net/cp-2019-69/cp-2019-69-AC2-supplement.pdf

[Figure]

LIA - CWP

[revised manuscript text omitted]

---

## Author Comment (AC3) · 19 Oct 2019

Figure S2

[Figure]

[Figure]

**Figure S2: Mass stream function during the LIA-CWP (shades), for austral summer (DJF), austral winter (JJA) and annual mean (ANN). for each model. Zonal mean precipitation (dash-dotted lines), zonal mean winds (solid lines), subtropical ridge position (lower stars) and latitude of maximum zonal winds at 200 hPa (upper stars) during the LIA (cyan) and the CWP (fuchsia) are also shown.**

**Fig. 1.**

---

## Author Comment (AC4) · 19 Oct 2019

Figure S2

/tmp/1818031900/figure-1.pdf

**Fig. 1.**

---

## Author Response (AR2)

Dear Editor,

We would like to thank you again for your suggestions. Here, we address new a point-to-point responses to comments.

All suggested changes to wording and punctuation have been incorporated.

On behalf of all co-authors,

Valentina Flores-Aqueveque

**Answer to Editor**

Figure 3: Is it needed? Isn't that the same information as show in Figure 1?

R: We integrated information of Figures 1 and 3 in a new version of Figure 1, removing previous Figure 3.

P16, L. 6: Figure 3 does not show 21C projections.

R: Yes, it was a mistake. This phrase (inherited from a previous version of the manuscript) was deleted.

Figure 4: Both reviewers were a bit critical of that figure, and I think it is still a bit difficult to read. The change between LIA and CDW is not obvious, particularly with respect to the stars.

R: Model simulations show almost no changes in SPSH center position. In addition, its horizontal resolution prevents further recognition of these small changes. For an easy viewing, we replaced stars with crosses.

Since, in the models, the meridional wind is only significant on the eastern and western sides of the SPSH (i.e. the winds are mostly zonal elsewhere), this figure can sometimes be hard to interpret, or maybe not appropriate. Would it make more sense to show wind as vectors instead?

R: To clarify results, we included vectors in Figure 3 (and the new Figure 4).

Adding the 21C data on top while the shading shows LIA-CDW is questionable. Would it make more sense to make an additional figure with only the 21C data (or 21C as anomalies from CDW)?

R: To avoid confusions, we included an additional figure (Figure 4) showing the difference between present-day (CWP) and future (21C) conditions.

Should the domain be extended to the west as the SPSH is "cut" in some of the plots?

R: In some cases of models CCSM4, IPSL-CM5A-LR and MRI-CGCM3, the contours used in plots to represent SPSH extend further west. However, this is a problem of the model output and this does not mean that the SPSH expands in that direction. Extend the domain to the west may produce a misconception of the extent of anticyclone.

I also wonder if the description in the text is accurate: P17, L.28-29: "The SPSH expansion during the CWP period is accompanied with weaker meridional winds in front of northern and central Chilean coast at annual

scale and during austral summer." It seems to me like there is a dipole pattern for most models, with stronger northward winds on the southeast side of the SPSH (~30S-35S), and weaker winds north of that.

R: The sentence was rephrased as: "The SPSH expansion during the CWP period is accompanied with stronger (weaker) southerly winds south (north) of ~35ºS on the southeastern side of the anticyclone, during the austral summer".

Figure 5: The stars are a bit small and hard to see, and the upper stars are missing in DJF. Please amend the legend to "CWP" for magenta.

R: As in Figure 4, we replaced the stars with crosses to help visualization. The legend was corrected.

In this work, the STJ is defined as the 'latitudinal position of the maximum zonal wind at 200 hPa, north of 48ºS` (P5, L31-32 of this updated version). On the other hand, the STJ weakens during austral summer (DJF). Therefore, is possible that, under our definition, the STJ does not reach the threshold velocity during austral summer north of 48ºS and that could be the reason why the upper stars are systematically absent in DJF (see Fig. 2S).

Minor Points:

P18, L.12: change to "models fail in representing the position"

R: Change was made.

P19, L. 16: change to "the numerical simulations analyzed here…"

R: Change was made.

P19, L. 20: "A comparison of our results with these previous studies.."

R: Change was made.

P19, L. 20: "that the models we have considered here represent…"

R: Change was made.

P19, L. 21-22: Please reformulate that sentence: "However, simulations are not in agreement with the position of STJ and its variations. "

R: The sentence was reformulated as: "However, simulations fail in representing the position of STJ and its variations".

P20, L.8-9: "Generally, a similar feature is observed in all models, but with different magnitude (Fig. 2S). "

R: Change was made.

P22, L.28: Please remove the comma after "results"

R: The comma was deleted.

[revised manuscript text omitted]